# Regioselective Claisen–Schmidt Adduct of 2-Undecanone from *Houttuynia cordata* Thunb as Insecticide/Repellent against *Solenopsis invicta* and Repositioning Plant Fungicides against *Colletotrichum fragariae*

**DOI:** 10.3390/molecules28166100

**Published:** 2023-08-17

**Authors:** Aigerim Kurmanbayeva, Meirambek Ospanov, Prabin Tamang, Farhan Mahmood Shah, Abbas Ali, Zeyad M. A. Ibrahim, Charles L. Cantrell, Satmbekova Dinara, Ubaidilla Datkhayev, Ikhlas A. Khan, Mohamed A. Ibrahim

**Affiliations:** 1National Center for Natural Products Research, School of Pharmacy, University of Mississippi, University, MS 38677, USA; aigerimkurmanbaeva14@gmail.com (A.K.); mospanov@olemiss.edu (M.O.); fshah@olemiss.edu (F.M.S.); aali@olemiss.edu (A.A.); ikhan@olemiss.edu (I.A.K.); 2Department of Organization, Management and Economics of Pharmacy and Clinical Pharmacy, School of Pharmacy, S.D. Asfendiyarov Kazakh National Medical University, Almaty 050012, Kazakhstan; dskanatovna@gmail.com; 3USDA-ARS, Natural Products Utilization Research Unit, University, MS 38677, USA; prabin.tamang@usda.gov (P.T.); charles.cantrell@usda.gov (C.L.C.); 4General Studies, University of Mississippi, University, MS 38677, USA; zmibrahi@go.olemiss.edu; 5The National Academy of Science, Republic of Kazakhstan, Almaty 050010, Kazakhstan

**Keywords:** *Houttuynia cordata* Thunb, Claisen condensation, 2-undecanone, *Colletotrichum fragariae*, *Solenopsis invicta*

## Abstract

The U.S. Department of Agriculture (USDA) has established research programs to fight the phytopathogen *Colletotrichum fragariae* and the invasive red imported fire ant, *Solenopsis invicta*. *C. fragariae* is known to cause anthracnose disease in fruits and vegetables, while *S. invicta* is known for its aggressive behavior and painful stings and for being the cause of significant damage to crops, as well as harm to humans and animals. Many plants have been studied for potential activity against *C. fragariae* and *S. invicta*. Among the studied plants, *Houttuynia cordata* Thunb has been shown to contain 2-undecanone, which h is known for its antifungal activity against *Colletotrichum gloesporioides*. Based on the mean amount of sand removed, 2-undecanone showed significant repellency at 62.5 µg/g, similar to DEET (*N*,*N*-diethyl-*meta*-toluamide), against *S. invicta*. The 2-Undecanone with an LC_50_ of 44.59 µg/g showed toxicity against *S. invicta* workers. However, neither *H. cordata* extract nor 2-undecanone had shown activity against *C. fragariae* despite their known activity against *C. gloesporioides*, which in turn motivates us in repositioning 2-undecanone as a selected candidate for a Claisen–Schmidt condensation that enables access to several analogs (**2a**–**f**). Among the prepared analogs, (*E*)-1-(3-methylbenzo[b]thiophen-2-yl)dodec-1-en-3-one (**2b**) and (*E*)-1-(5-bromothiophen-2-yl)dodec-1-en-3-one (**2f**) showed promising activity against *C. fragariae*, revealing a distinctive structural activity relationship (SAR). The generated analogs revealed a clear regioselectivity pattern through forming the C=C alkene bond at position C-1. These data open the window for further lead optimization and product development in the context of managing *C. fragariae* and *S. invicta*.

## 1. Introduction

The U.S. Department of Agriculture (USDA) is actively engaged in ongoing research programs to control the spread of the aggressive red imported fire ant (RIFA) and to find alternative treatments for the fungal phytopathogen, *Colletotrichum fragariae,* which is known to cause anthracnose disease in crucial crops such as strawberries.

Red imported fire ants (*Solenopsis invicta* Buren; RIFA) (Hymenoptera: Formicidae), native to South America, are serious exotic pests of quarantine significance [1]. Increasing populations of imported fire ants negatively affect humans, plants, wildlife, and agriculture through their aggressive stinging and digging behavior. Ecological impacts include predation, competition with native ant species, and biodiversity loss leading to changes in ecosystem processes [2,3,4,5,6,7]. Toxic baits and synthetic contact insecticides are the most used form of imported fire ant control. Due to the repeated, long-term use of these methods, fire ants have acquired behavioral and physiological mechanisms to counter synthetic toxicants leading to pest control complications [8,9,10,11,12,13]. Moreover, the use of conventional insecticides is considered inappropriate in and around nursing homes, hospitals, laboratories, stock, circuitry, and storage [14,15]. There is a federal quarantine law in effect in the United States to prevent fire ants from spreading to non-infested areas (http://www.aphis.usda.gov/ppq/ispm/fireants/index.html (accessed on 29 October 2022)). Plant-derived natural toxicants and repellents may offer novel alternative and effective means of managing populations of imported fire ants.

*C. fragariae* is a fungal pathogen that belongs to the genus *Colletotrichum* and is known to cause anthracnose disease in various fruits and vegetables including strawberries, significantly impacting strawberry production in the United States [16,17]. Anthracnose disease is more prevalent in humid and warm areas, such as the Southeastern United States, where strawberry production is a major agricultural industry [18]. This disease can spread rapidly in the field, especially under favorable environmental conditions, making it challenging to control [18]. Farmers typically use a combination of cultural practices, such as crop rotation, sanitation, resistant cultivars, and fungicides [19]. Among these management practices, chemical fungicides are commonly used for managing this disease, but their effectiveness is challenged by the increasing fungicide resistance of the pathogens. Furthermore, chemical fungicides have detrimental impacts on human health, the environment, and non-targeted beneficial organisms [20]. As a result, there is an urgent need to find alternative strategies for the management of this disease. Research is ongoing to explore and develop new strategies for controlling *C. fragariae*, including the utilization of biocontrol agents and the development of resistant cultivars [20]. Research is ongoing to understand the biology and genetics of the pathogen, as well as its interactions with the strawberry plant and the environment in order to develop effective disease management strategies [21]. Despite these efforts, *C. fragariae* continues to pose a significant challenge for strawberry production in the United States and remains a major focus of research led by the USDA-ARS and the land grant universities.

Many plants have been evaluated for their potential insect repellency and fungicidal activities against *S. invicta* and against *C. fragariae*, respectively. Among the studied plants, *Houttuynia cordata* Thunb, also known as the “Chameleon plant” or “Fish mint”, is an herbaceous perennial plant belonging to the Saururaceae family and is native to East Asia [22]. *H. cordata* has been traditionally used in Chinese and Japanese medicines for its anti-inflammatory, antiviral, and antibacterial properties [23,24]. Also, it has been used as a food supplement and as an active ingredient in beauty care cosmetics in these countries [25].

One of the most interesting secondary metabolites identified from *H. cordata* is 2-undecanone (methyl nonyl ketone, MNK) [26]. It is mostly used as an insect repellant, particularly for mosquitoes, due to its strong odor [27]. It has been formulated in various forms such as liquid, aerosol spray, and gels in commercial applications and used as insect repellents for dogs and cats in concentrations ranging from 1–2% [28]. Recent studies have shown its activity against the fungal pathogen, *C. gloesporioides* [28].

In the present study, we evaluated *H. cordata* extract and 2-undecanone against *S. invicta* and *C. fragariae* and used the rationale repositioning of 2-undecanone as a selected candidate for the Claisen–Schmidt condensation to enable access to several analogs (**2a**–**f**) for further evaluation against *C. fragariae.*

## 2. Results

### 2.1. Digging Bioassay

The mean weight (g) of treated sand removed by red imported fire ant workers in a digging bioassay with different concentrations of 2-undecanone and DEET is presented in Figure 1. Repellency was determined based on the minimum repellent effective dose (µg/g) where the quantity of sand removed was similar to that of the ethanol control. Based on the amount of sand removed, 2-undecanone (*F*_3,8_ = 13.36; *p* = 0.002; Figure 1a) and DEET (F_3,8_ = 16.24; *p* < 0.001; Figure 1b), both showed a significantly higher repellency than ethanol at dosages of 125 and 62.5 µg/g, whereas the activity at 31.25 µg/g was similar to ethanol.

### 2.2. Toxicity Bioassay

Toxicity data of 2-undecanone and bifenthrin at 24 h post treatment against RIFA workers are given in Table 1. 2-Undecanone with LC_50_ values of 44.59 µg/g showed toxicity against red imported fire ant workers. Bifenthrin with an LC_50_ value of 0.03 µg/g was more toxic against RIFA than 2-undecanone.

### 2.3. Chemistry

2-Undecanone is known for its fungal activity against *C. gloesporioides*, however, neither *H. cordata* extract nor 2-undecanone had shown activity against *C. fragariae*, which motivated us in repositioning 2-undecanone as a selected candidate for the Claisen–Schmidt condensation (Figure 2).

### 2.4. Direct Antifungal Bioautography Assay

A direct bioautography assay helps to determine the potential of 2-undecanone and its analogs as antifungal agents against anthracnose disease in fruits and vegetables, and it provides valuable information for further studies on their mechanism of action and potential for agricultural applications. Compound **2b** showed zones of inhibition of 0.5, 0.91, and 1.2 cm at the tested concentrations of 5, 10, and 15 µL of 10.0 mg/mL prepared stock solution, respectively, while Compound **2f** showed zones of inhibition of 0.7, 1.0, and 1.5 cm at the tested concentrations of 5, 10, and 15 µL of 10.0 mg/mL prepared stock solution, respectively (Table 2).

## 3. Discussion

Insecticidal, repellent, and behavioral effects of 2-undecanone have been well established [29,30]. Whang and Tonelli [31] reported that 2-undecanone from its crystalline inclusion compound with α-cyclodextrin repelled 100% of cockroaches for the first 2 days after application. Ali et al. [32] reported that the biting deterrent activity against *Aedes aegypti* L. of 2-undecanone at 8.5 μg/cm^2^ was similar to DEET at 4.8 μg/cm^2^. Ntalli et al. [33] reported that 2-undecanone caused 87.8% and 98.9% mortality of the larvae and adults of *Tenebrio molitor* L., respectively, at 7 days post exposure to wheat treated at a dose of 1000 μL/kg. Data from the present study demonstrated that 2-undecanone significantly suppressed ant digging activity. It repelled the ants, and the minimum effective dose of repellency was 62.5 µg/g below which the repellency of this natural product failed. The level of ant activity in this treatment and the DEET treatment were similar, suggesting that 2-undecanone can offer promise against the red imported fire ant. The toxicity data from the present study showed that 2-undecanone was toxic to the red imported fire ants. Taken together, our data further support the idea that 2-undecanone can be a promising pest management tool. The Claisen–Schmidt condensation is a valuable chemistry tool invented by Rainer Ludwig Claisen and J. G. Schmidt [34]. The reaction is between an aldehyde or ketone having an α-hydrogen with an aromatic carbonyl compound lacking an α-hydrogen [34]. An example is the synthesis of dibenzylideneacetone (1*E*,4*E*)-1,5-diphenylpenta-1,4-dien-3-one) [35], where dibenzylideneacetone is used as a component in sunscreens and as a ligand in organometallic chemistry [36]. Using our lead 2-undecanone as a ketone model enables us to generate a library of six analogs. The reaction in general was of a reasonable yield for most of the presented aromatic aldehydes, therefore, the feasibility and the success rate are quite high as a practical approach (Figure 2). As predicted, with respect to regioselectivity, our products are mainly formed at the 1-position. The model molecules have been examined through ChemDraw 21.0 for their Log P and pKa, where they showed a Log P of ~2.5–3.5 and pKa of ~8.5–9.5 confirming their favorable drug-like properties. Also, they have shown considerable solubility in the assay aqueous condition at a maximum concentration of 20 µM. Among the prepared analogs, **2b** and **2f** have shown increased activity against *C. fragariae* in comparison with the parent molecule **1,** which did not show any antifungal activity against *C. fragariae*. The two active analogs of 2-undecanone, (*E*)-1-(3-methylbenzo[b]thiophen-2-yl)dodec-1-en-3-one (**2b**) and (*E*)-1-(5-bromothiophen-2-yl)dodec-1-en-3-one (**2f**), demonstrated faint clear zones on the TLC plate, which in turn highlighted the importance of such reactions in the development of fungicidal products. Moreover, compounds **2a**–**b** showed a distinctive SAR, where the introduction of a methyl group at C-3 of the benzothiophene ring (**2b**) is a promoter for increasing activity against *C. fragariae* in comparison to the unsubstituted derivative (**2a**).

## 4. Experimental Section

### 4.1. General Experimental Procedures

The ^1^H and ^13^C-NMR spectra were recorded in DMSO-*d*_6_ and CDCl_3_ on a Bruker 400 and 500 MHz spectrometer operating at 400, 500 MHz for ^1^H and 100, 150 MHz for ^13^C NMR. Chemical shift (*δ*) values are presented in ppm and in reference to the residual solvent signals of DMSO-*d*_6_ and CDCl_3_ at *δ*_H_/*δ*_C_ 2.50/39.5 and 7.25/70.2, respectively. The coupling constant (*J*) is reported in Hz.

The LC analysis of the generated analogs (**2a**–**2f**) was conducted using an Agilent 1100 HPLC system, RP-C18 column (150 × 4.6 mm; particle size of 5 μm; Luna) with column oven temperature set at 25 °C, and a gradient system of eluent water (A) and acetonitrile (B). The gradient condition was as follows: 0–2 min (5% B), 2–5 min (5% B → 50% B), 5–10 min (50% B → 100% B), and 10–15 min (100% B). The flow rate of the solvent was 1.0 mL/min, and the injection volume was 25 μL. All analyses were carried out at a wavelength of 254 nm with a run time of 15 min. HPLC-grade acetonitrile and water solvents were used. Acetic acid was added as a modifier to achieve a final concentration of 0.1% in each solvent.

Preparative HPLC purification of the generated analogs (**2a**–**2f**) was carried out using an Agilent 1100 HPLC system, RP-C18 column (250 × 10 mm; particle size of 10 μm; Luna) with column oven temperature set at 25 °C and a gradient system of eluent water (A) and acetonitrile (B). The gradient condition was as follows: 0–2 min (5% B), 2–5 min (5% B → 50% B), 5–10 min (50% B → 100% B), and 10–15 min (100% B). The flow rate of the solvent was 3.0 mL/min, and the injection volume was 25 μL. All analyses were carried out at a wavelength of 254 nm with a run time of 15 min. HPLC-grade acetonitrile and water solvents were used. Acetic acid was added as a modifier to achieve a final concentration of 0.1% in each solvent.

Other common chromatographic techniques, such as thin layer chromatography (TLC) on precoated silica gel G_254_ aluminum plates and silica gel flash column chromatography, were also engaged in the purification of the synthesized compounds.

### 4.2. General Method for Synthesis of (E)-1-(Arylmethylene)dodec-1-en-3-one (2a-2f)

The 2-Undecanone (1.0 mmol), (Appendix A), was added to a solution of aldehyde (25 mg, 1.0 mmol) in methanol (5 mL). One mL of 20% NaOH was added, and the solution refluxed. Various modifications were used to obtain compounds **2a**–**2f**. Benzothiophene-2-carboxaldehyde, benzothiophene-3-carboxaldehyde, and indole-5-carboxaldehyde were purchased from TCI AMERICA (Portland, OR, USA) with purity ≥98%. 3-Methylbenzothiophene-2-carboxaldehyde and 5-bromothiophene-2-carboxaldehyde were purchased from Alfa Aesar (Ward Hill, MA, USA) with purity ≥98% and ≥98%, respectively. 1-methylindole-3-carboxaldehyde was purchased from ACROS ORGANICS (Morris Plains, NJ, USA) with purity ≥97%.

#### 4.2.1. (*E*)-1-(Benzo[b]thiophen-2-yl)dodec-1-en-3-one (**2a**)

The starting material benzothiophene-2-carboxaldehyde (25 mg, 0.15 mmol) was used, and the reaction mixture was stirred by refluxing overnight for 15 h. Neutralization was performed using HCl, filtered, and dried to afford a yellow solid of **2a**. Yield of 35 mg (72.0%); ^1^H-NMR (400 MHz, CDCl_3_): δ = 0.90 (t, *J* = 6.8 Hz, 3H, H-12), 1.32 (m, 12H, H-6, H-7, H-8, H-9, H-10, and H-11), 1.70 (m, 2H, H-5), 2.66 (t, *J* = 7.4 Hz, 2H, H-4), 6.62 (d, *J* = 15.7 Hz, 1H, H-2), 7.39 (m, 2H, H-5′ and H-6′), 7.52 (s, 1H and H-3′), and 7.79 (m, 3H, H-1, H-4′, and H-7′); ^13^C-NMR (100 MHz, CDCl_3_): δ = 14.13 (C-12), 22.69 (C-11), 24.33 (C-5), 29.30 (C-6), 29.35 (C-7 and C-9), 29.48 (C-8), 31.90 (C-10), 41.35 (C-4), 122.49 (C-3′), 124.41 (C-4′), 124.91 (C-7′), 126.33 (C-5′), 127.14 (C-6′), 129.30 (C-1), 135.18 (C-2), 139.66 (C-3a’), 139.99 (C-7a’), 140.19 (C-2′), and 199.98 (C=O); HRMS m/z calcd. for C_20_H_27_OS [M + H]^+^ 315.17826, found 315.17826; (Appendix A).

#### 4.2.2. (*E*)-1-(3-Methylbenzo[b]thiophen-2-yl)dodec-1-en-3-one (**2b**)

The starting material 3-methylbenzothiophene-2-carboxaldehyde (25 mg, 0.14 mmol) was used, and the reaction mixture was stirred by refluxing overnight for 15 h. Neutralization was performed using HCl, filtered off, and dried to afford a yellow solid of **2b**. Yield of 33 mg (70.0%); ^1^H-NMR (400 MHz, CDCl_3_): δ = 0.91 (bt, 3H, H-12), 1.32 (m, 12H, H-6, H-7, H-8, H-9, H-10, and H-11), 1.71 (m, 2H, H-5), 2.55 (s, 3H, H-8), 2.66 (t, *J* = 7.4 Hz, 2H, H-4), 6.62 (d, *J* = 15.5 Hz, 1H, H-2), 7.41 (m, 2H, H-5′ and H-6′), 7.73 (m, 1H, H-4′), 7.79 (m, 1H, H-7′), and 7.96 (d, *J* = 15.5 Hz, 1H, H-1); ^13^C-NMR (100 MHz, CDCl_3_): δ = 12.22 (C-8′), 14.13 (C-12), 22.69 (C-11), 24.40 (C-5), 29.31 (C-8), 29.36 (C-6), 29.48 (C-7 and C-9), 31.90 (C-10), 41.89 (C-4), 122.46 (C-4′), 124.55 (C-7′), 126.18 (C-5′), 126.64 (C-2), 133.11 (C-1), 134.13 (C-3′), 137.17 (C-6′), 139.17 (C-7a’), 140.68 (C-3a’), and 200.03 (C=O); HRMS m/z calcd. for C_21_H_29_OS [M + H]^+^ 329.19391, found 329.19391; (Appendix A).

#### 4.2.3. (*E*)-1-(Benzo[b]thiophen-3-yl)dodec-1-en-3-one (**2c**)

The starting material benzothiophene-3-carboxaldehyde (25 mg, 0.15 mmol) was used, and the reaction mixture was stirred by refluxing overnight for 15 h. Neutralization was performed using HCl, filtered off, and dried to afford a yellow solid of **2c**. Yield of 36 mg (75.0%); ^1^H-NMR (400 MHz, CDCl_3_): δ = 0.90 (bt, 3H, H-12), 1.29 (m, 12H, H-6, H-7, H-8, H-9, H-10, and H-11), 1.73 (m, 2H, H-5), 2.71 (t, *J* = 7.4 Hz, 2H, H-4), 6.62 (d, *J* = 16.1 Hz, 1H, H-2), 7.47 (m, 2H, H-5′ and H-6′), 7.82 (s, 1H, H-2′), 7.88 (d, *J* = 16.2 Hz, 1H, H-1), 7.92 (d, *J* = 7.9 Hz, 1H, H-7′), and 8.05 (d, *J* = 8.0 Hz, 1H, H-4′); ^13^C-NMR (100 MHz, CDCl_3_): δ = 14.12 (C-12), 22.69 (C-11), 24.47 (C-5), 29.31 (C-6), 29.39 (C-8), 29.49 (C-7 and C-9), 31.90 (C-10), 41.30 (C-4), 122.07 (C-2′), 123.09 (C-7′), 124.98 (C-4′), 125.12 (C-5′), 126.61 (C-6′), 128.18 (C-2), 131.66 (C-3′), 133.83 (C-7a’), 137.28 (C-1), 140.59 (C-3a’), and 200.61 (C=O); HRMS m/z calcd. for C_20_H_27_OS [M + H]^+^ 315.17826, found 315.17826; (Appendix A).

#### 4.2.4. (*E*)-1-(1H-Indol-5-yl)dodec-1-en-3-one (**2d**)

The starting material indole-5-carboxaldehyde (25 mg, 0.17 mmol) was used, and the reaction mixture was stirred by refluxing overnight for 15 h. Neutralization was performed using HCl, filtered off, and dried to afford a yellow solid of **2d**. Yield of 34 mg (67.0%); ^1^H-NMR (400 MHz, CDCl_3_): δ = 0.90 (bt, 3H, H-12), 1.29 (m, 12H, H-6, H-7, H-8, H-9, H-10, and H-11), 1.72 (m, 2H, H-5), 2.69 (t, *J* = 7.4 Hz, 2H, H-4), 6.61 (s, 1H, NH), 6.77 (d, *J* = 16.1 Hz, 1H, H-2), 7.27 (m, 2H, H-2′ and H-3′), 7.45 (m, 2H, H-6′ and H-7′), 7.74 (d, *J* = 16.1 Hz, 1H, H-1), and 7.86 (s, 1H, H-4); ^13^C-NMR (100 MHz, CDCl_3_): δ = 14.13 (C-12), 22.69 (C-11), 23.90 (C-5), 24.72 (C-8), 29.45 (C-6), 29.50 (C-7 and C-9), 31.90 (C-10), 40.89 (C-4), 103.51 (C-3′), 111.67 (C-7′), 121.75 (C-4′), 122.77 (C-6′), 123.77 (C-2′), 125.38 (C-2), 126.64 (C-3a’), 128.26 (C-7a’), 137.16 (C-5′), 144.47 (C-1), and 201.13 (C=O); HRMS m/z calcd. for C_20_H_28_NO [M + H]^+^ 298.21709, found 298.21709; (Appendix A).

#### 4.2.5. (*E*)-1-(1-Methyl-1H-indol-3-yl)dodec-1-en-3-one (**2e**)

The starting material 1-methylindole-3-carboxaldehyde (25 mg, 0.16 mmol) was used, and the reaction mixture was stirred by refluxing overnight for 15 h. Neutralization was performed using HCl, filtered off, and dried to afford a yellow solid of **2e**. Yield of 35 mg (71.0%); ^1^H-NMR (400 MHz, CDCl_3_): δ = 0.91 (bt, 3H, H-12), 1.33 (m, 12H, H-6, H-7, H-8, H-9, H-10, and H-11), 1.73 (m, 2H, H-5), 2.66 (t, *J* = 7.4 Hz, 2H, H-4), 3.84 (s, 3H, H-8), 6.80 (d, *J* = 16.0 Hz, 1H, H-2), 7.34 (m, 3H, H-7, H-6, and H-5), 7.40 (s, 1H, H-2), 7.82 (d, *J* = 16.0 Hz, 1H, H-1), and 7.96 (m, 1H, H-4); ^13^C-NMR (100 MHz, CDCl_3_): δ = 14.12 (C-12), 22.70 (C-11), 24.96 (C-5), 29.33 (C-8), 29.52 (C-6), 25.54 (C-7 and C-9), 31.91 (C-10), 33.25 (C-8′), 40.99 (C-4), 110.04 (C-7′), 112.26 (C-3′), 120.65 (C-4′), 121.36 (C-5′), 121.66 (C-6′), 123.08 (C-3a’), 133.69 (C-2), 135.97 (C-7a’), 138.19 (C-1), and 200.99 (C=O); HRMS m/z calcd. for C_21_H_30_NO [M + H]^+^ 312.23274, found 312.23274; (Appendix A).

#### 4.2.6. (*E*)-1-(5-Bromothiophen-2-yl)dodec-1-en-3-one (**2f**)

The starting material 5-bromothiophene-2-carboxaldehyde (25 mg, 0.13 mmol) was used, and the reaction mixture was stirred by refluxing overnight for 15 h. Neutralization was performed using HCl, filtered off, and dried to afford a yellow solid of **2f**. Yield of 35 mg (78.0%); ^1^H-NMR (400 MHz, CDCl_3_): δ =0.89 (bt, 3H, H-12), 1.29 (m, 12H, H-6, H-7, H-8, H-9, H-10, and H-11), 1.66 (m, 2H, H-5), 2.59 (t, *J* = 7.4 Hz, 2H, H-4), 6.44 (d, *J* = 15.8 Hz, 1H, H-2), 7.03 (s, 2H, H-3 and H-4), and 7.54 (d, *J* = 15.8 Hz, 1H, H-1); ^13^C-NMR (100 MHz, CDCl_3_): δ = 14.12 (C-12), 22.65 (C-11), 24.33 (C-5), 29.29 (C-8), 29.31 (C-6), 29.46 (C-7 and C-9), 31.88 (C-10), 41.36 (C-4), 116.08 (C-5′), 125.09 (C-4′), 131.20 (C-1), 131.76 (C-2), 133.75 (C-3′), 141.57 (C-2′), and 199.76 (C=O); HRMS m/z calcd. For C_16_H_24_BrOS [M + H]^+^ 343.07312, found 343.07312; (Appendix A).

### 4.3. Imported Fire Ants

RIFA used in these bioassays were collected from Washington County, MS 38748 (33°09′31.2″ N 90°54′56.4″ W). A plastic tray was used to contain the colony, and the tray walls were coated with Insect-a-Slip (BioQuip Products, 2321 Gladwick Street, Rancho Dominguez, CA 90220, USA) to prevent the fire ants from crawling out. A 25% solution of water and sugar was provided to the ant colonies, and crickets were added weekly as food sources. A water-filled test tube plugged with cotton was added to serve as a moisture source. The laboratory conditions were set at 27 ± 2 °C and 60 ± 10% RH with a photoperiod regimen of 12:12 h (L:D). Ants were maintained under this setting for one month before starting the bioassays. The venom alkaloid and hydrocarbon indices of the workers were used for the identification of the imported fire ant species [37,38].

#### 4.3.1. Digging Bioassay

The digging bioassay used in this study for testing repellency was described by Ali et al. [37]. Briefly, the bioassay consisted of a 150 mm × 15 mm Petri dish (Fisher Scientific Co. LLC, 2775 Horizon Ridge CT, Suwanee, GA 30024, USA) and four 2 mL Nylgene Cryoware Cryogenic vials (Thermo Fisher Scientific, Rochester, NY 14825, USA), which were glued to the lower side of the Petri dish. The inner walls of the Petri dish were coated with Insect-a-Slip (BioQuip Products, 2321 Gladwick Street, Rancho Dominguez, CA 90220, USA). Four grams of sand (Premium Play Sand, Plassein International, Longview, TX, USA) of uniform size (500 microns) were weighed in fluted aluminum (45 mL size) dishes (Fisher Scientific, 300 Industry Drive, Pittsburgh, PA 15275, USA), and each treatment was added at 100 µL/g of sand. The control sand received treatment with ethanol only. After the solvent had evaporated, the sand was moistened by adding 0.6 µL/g of de-ionized water and was filled into the treatment vials, which were then screwed to the Petri dish. Each vial contained a mean weight of 3.6 g of sand on a dry weight basis. Fifty imported fire ant workers were released per Petri dish. The experiments were conducted at 25 ± 2 °C temperature and 50 ± 10% relative humidity for a 12 h (L:D) photoperiod. After 24 h, sand from the treatment vials was collected back into aluminum dishes, oven dried at 190 °C for 1 h, and weighed. Both 2-undecanone and DEET (*N*,*N*-diethyl-meta-toluamide) were tested. DEET was tested as a positive control for comparison whereas ethanol was included as a negative control. Serial doses were started from a dose of 125 µg/g until the failure of the treatment. The stock solution and concentrations were prepared in ethanol. Overall, three sets of experimental replicates were run on 3 different days. Sand removal data were analyzed using Analysis of Variance (ANOVA) and means were separated using Ryan–Einot–Gabriel–Welsch multiple range test (*p* ≤ 0.05; SPSS, ver. 25), (Figure 1).

#### 4.3.2. Toxicity Bioassay

The toxicity bioassay used for determining mortality in imported fire ants was described by Ali et al. [37]. Briefly, 3 g of sand was weighed into fluted 42 mL aluminum dishes and treated with 300 µL of treatment solution. Control sand was treated with ethanol only. The solvent was evaporated, and 0.6 µL/g of de-ionized water was added to moisten the sand. Treatment sands were transferred into 60 × 15 mm stackable Petri dishes (KORD-VALMARK, Mfg by Bioplast Manufacturing, L.L.C. 128 Wharton Road, Bristol, PA 19007, USA). Ten fire ant workers were released per treatment/replicate. A wet cotton swab tip was added to the Petri dish 1 h post release of the workers to ensure continuous availability of the moisture. The number of dead workers was recorded at 24 h post treatment. LC_50_ values were calculated using probit analysis (SAS 2012), (Table 1).

### 4.4. Direct Bioautography Assay

The direct bioautography assay was conducted to assess the antifungal activity of 2-undecanone and its six analogs against the phytopathogen *C. fragariae*. First, the inoculum of *C. fragariae* was prepared by harvesting fresh conidia from a 7–10-day old fungal culture grown on ½ strength potato dextrose agar (PDA). The conidia were collected by flooding the plate with 10 mL of sterile water and scrapping it with a sterile L-shaped spatula. The resulting spore suspension was filtered through a sterile double Mira cloth (Calbiochem-Novabiochem Corp., La Jolla, CA, USA) to remove mycelia and agar pieces, and the spores were counted using a Countess 3 cell counter. The spore suspension was centrifuged at 1968 RCF for 10 min, and the supernatant was discarded. The spore concentration was then adjusted to 3 × 10^5^ spores mL^−1^ by adding the required volume of PDB-TLC media (12.5 g PDB, 0.5 g agar, 0.5 mL Tween 80 in 500 mL of water) to obtain the inoculum for the bioautography assay.

Subsequently, 10 µL of each compound (2-undecanone and its six analogs **2a**–**f**) was spotted twice onto a silica gel plate (Analtech, Inc., Newark, DE, USA, Silica Gel GHLF, 250 microns). The inoculum of *C. fragariae* was then uniformly sprayed onto the silica gel plate using a hand sprayer, ensuring that the spores were evenly distributed across the plate. The TLC plate was placed in a moisture chamber box with 99.9% humidity and incubated at approximately 26 °C for 4 days, which provides optimal conditions for the growth of *C. fragariae*. After incubation, the antifungal activity of the compounds was evaluated by the presence of clear zones on the TLC plate, which indicates the inhibition of fungal growth. Captan (>98%, Chem Service, Inc., West Chester, PA, USA) and fludioxonil (>99.5%, Chem Service, Inc., West Chester, PA, USA), which are technical-grade fungicides, were used as positive controls at a concentration of 2 mg/mL in ethanol, with 1 µL of each control applied on the plate to compare the antifungal activity of the test compounds.

## 5. Conclusions

*H. cordata* Thunb, has been shown to contain 2-undecanone, which is known for its fungal activity against *C. gloesporioides*; however, it does not possess activity against *C. fragariae.* Repositioning 2-undecanone via a Claisen–Schmidt condensation enables access to six analogs (**2a**–**f**). Among these analogs, (*E*)-1-(3-methylbenzo[b]thiophen-2-yl)dodec-1-en-3-one (**2b**) and (*E*)-1-(5-bromothiophen-2-yl)dodec-1-en-3-one (**2f**) showed promising activity against *C. fragariae*, revealing an interesting SAR and clear regioselectivity pattern through the formation of the C=C alkene bond at position C-1. Alongside that, 2-Undecanone significantly suppressed the digging activity at 62.5 µg/g, similar to DEET, against red imported fire ants, and it showed toxicity with an LC_50_ of 44.59 µg/g against fire ant workers. These data suggested that 2-undecanone may play a useful role in the integrated management of *C. fragariae* and *S. invicta*.

## Figures and Tables

**Figure 1 molecules-28-06100-f001:**
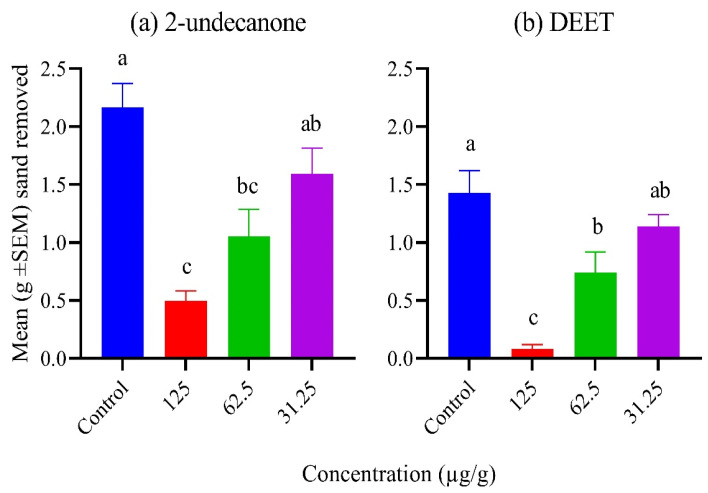
Mean weights (g) of treated sand removed by the workers of red imported fire ant, released in digging bioassays, with different concentrations of 2-undecanone and DEET. Means within each experiment, not followed by the same letter, are significantly different (Ryan–Einot–Gabriel–Welsch multiple range test, *p* ≤ 0.05).

**Figure 2 molecules-28-06100-f002:**
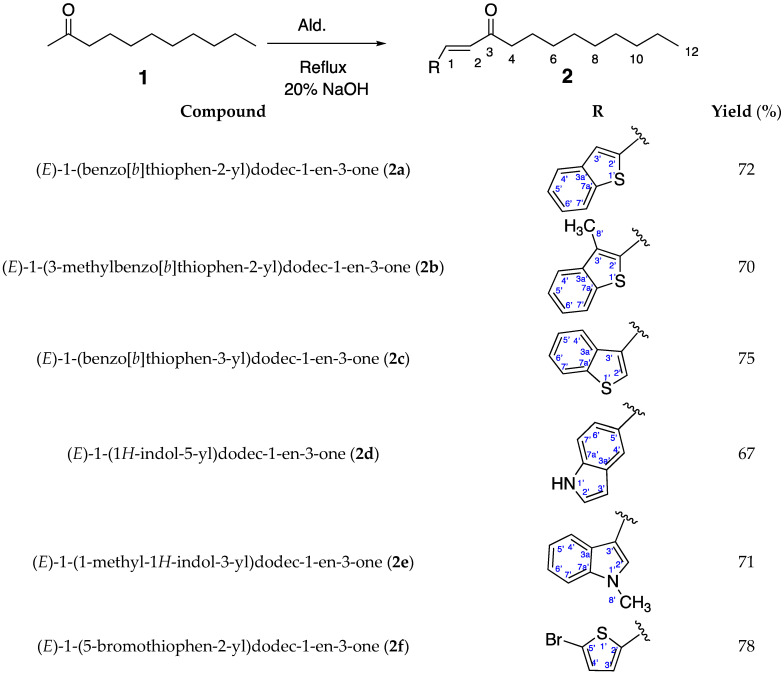
Reagents and conditions: 20% NaOH, reflux, overnight.

**Table 1 molecules-28-06100-t001:** Toxicity of 2-undecanone and bifenthrin against workers of red imported fire ants at 24 h post treatment.

Compound	n ^†^	Slope ± SE	LC_50_ (95% CI) ^‡^	LC_90_ (95% CI) ^‡^	χ^2^	*df*
2-Undecanone	30	2.32 ± 0.35	44.59 (38.32–52.09)	77.32 (64.17–104.31)	14.93	13
Bifenthrin	40	1.21 ± 0.18	0.03 (0.023 ± 0.04)	0.09 (0.06 ± 0.16)	42	19

^†^ n is the number of workers tested in each treatment. ^‡^ LC_50_ and LC_90_ values are in µg/g, and CIs are confidence intervals.

**Table 2 molecules-28-06100-t002:** Inhibition zones (cm) of the parent compound **1** and the generated analogs **2a**–**2f** against *C. fragariae* using 10 mg/mL stock solutions.

Cpds	Inhibition Zone Diameter (cm) against *C. fragariae* at 5 µL	Inhibition Zone Diameter (cm) against *C. fragariae* at 10 µL	Inhibition Zone Diameter (cm) against *C. fragariae* at 15 µL
**1**	-	-	-
**2a**	-	-	-
**2b**	0.5	0.91	1.2
**2c**	-	-	-
**2d**	-	-	-
**2e**	-	-	-
**2f**	0.7	1.0	1.5
Captan	1.8		
Fludioxonil	1.7		

(-) inactive up to the maximum measured concentration of 15 µL of 10.0 mg/mL prepared stock solutions. The captan and fludioxonil were 1 µL volume of 2 mg/mL stock solution.

## Data Availability

Data relevant to the paper will be available from the authors upon request.

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
