# Peer review of "Regioselective Claisen–Schmidt Adduct of 2-Undecanone from Houttuynia cordata Thunb as Insecticide/Repellent against Solenopsis invicta and Repositioning Plant Fungicides against Colletotrichum fragariae"

_molecules, 2023, doi:10.3390/molecules28166100_

Round 1

Reviewer 1 Report

It is an interesting study on a new antifungal and antiinsecticidal substancje of naturalna origin. It is very importand to look for new insect repellents and fungicides that would be safer for the environment than older generstios of such products. Fire ants and fungal plant diseases are a heavy burden for farmers thereforesuch such studies have a chance of practucal application of their results. 

Author Response

Reviewer 1:

Comment 1. It is an interesting study on a new antifungal and anti-insecticidal substances of natural origin. It is very important to look for new insect repellents and fungicides that would be safer for the environment than older generations of such products. Fire ants and fungal plant diseases are a heavy burden for farmers therefore such studies have a chance of practical application of their results. 

Response 1. We valued the respected reviewer’s appreciation of the presented work and indeed “Fire ants and fungal plant diseases are a heavy burden for farmers therefore such studies have a chance of practical application of their results”. 

Reviewer 2 Report

The manuscript entitled Regioselective Claisen-Schmidt Adduct of 2-Undecanone from Houttuynia cordata Thunb as Insecticides/Repellent against Solenopsis invicta and Repositioning Plant Fungicides against Colletotrichum fragariae reported some compounds that showed promising activity against C. fragariae,and the generated analogs revealed clear regioselectivity pattern through forming the C=C alkene bond at position C-1.  

The manuscript is well written and carefully described, while lacking the conclusion section. The authors should rewrite the conclusion and characterize the advantages of synthesized compounds in this paper with the existing reports.

Author Response

Reviewer 2:

The manuscript entitled “Regioselective Claisen-Schmidt Adduct of 2-Undecanone from Houttuynia cordata Thunb as Insecticides/Repellent against Solenopsis invicta and Repositioning Plant Fungicides against Colletotrichum fragariae” reported some compounds that showed promising activity against C. fragariae, and the generated analogs revealed clear regioselectivity pattern through forming the C=C alkene bond at position C-1.  

Comment 1. The manuscript is well written and carefully described, while lacking the conclusion section. The authors should rewrite the conclusion and characterize the advantages of synthesized compounds in this paper with the existing reports.

Response 1. We valued the respected reviewer’s recognition of the presented work and the conclusion part has been implemented as suggested.

Reviewer 3 Report

The authors report Regioselective Claisen-Schmidt Adduct of 2-Undecanone from Houttuynia cordata Thunb as Insecticides/Repellent against Solenopsis invicta and Repositioning Plant Fungicides against Colletotrichum fragariae.

C. fragariae is known to cause anthracnose disease in fruits and vegetables while S. invicta is known for its aggressive behavior and painful stings and for being the cause of significant damage to crops, as well as harm to humans and animals. Many plants have been studied for potential activity against C. fragariae and S. invicta. Among the studied plants, Houttuynia cordata Thunb, has been shown to contain 2-undecanone which is known for its antifungal activity against Colletotrichum gloesporioides. Based on the mean amount of sand removed, 2-undecanone showed significant repellency at 62.5 µg/g similar to DEET (N, N-diethyl-meta-toluamide) against S. invicta. 2-Undecanone with LC50 of 29 44.59 µg/g showed toxicity against S. invicta workers. However, neither H. cordata extract nor 2-undecanone had shown activity against C. fragariae despite its known activity against C. gloesporioides, which in turn motivates us in repositioning 2-undecanone as a selected candidate for Claisen-Schmidt condensation that enable access to several analogs (2a-f). Among the prepared analogs, (E)-1-(3-methylbenzo[b]thiophen-2-yl)dodec-1-en-3-one (2b) and (E)-1-(5-bromothiophen-2-yl)do- 34 dec-1-en-3-one (2f) showed promising activity against C. fragariae revealing a distinctive structural activity relationship (SAR). The generated analogs revealed a clear regioselectivity pattern through forming the C=C alkene bond at position C-1. These data open the window for further lead optimization and product development in the context of managing C. fragariae and S. invicta.

The work developed by the authors is very interesting, however, the study requires make minor and major corrections before being published. These corrections are shown in the attached document.

After making the indicated corrections, the article could be published in Molecules Journal.

Author Response

Reviewer 3:

The authors report Regioselective Claisen-Schmidt Adduct of 2-Undecanone from Houttuynia cordata Thunb as Insecticides/Repellent against Solenopsis invicta and Repositioning Plant Fungicides against Colletotrichum fragariae.

  1. fragariaeis known to cause anthracnose disease in fruits and vegetables while S. invictais known for its aggressive behavior and painful stings and for being the cause of significant damage to crops, as well as harm to humans and animals. Many plants have been studied for potential activity against C. fragariae and S. invicta. Among the studied plants, Houttuynia cordata Thunb, has been shown to contain 2-undecanone which is known for its antifungal activity against Colletotrichum gloesporioides. Based on the mean amount of sand removed, 2-undecanone showed significant repellency at 62.5 µg/g similar to DEET (N, N-diethyl-meta-toluamide) against S. invicta. 2-Undecanone with LC50 of 29 44.59 µg/g showed toxicity against S. invicta workers. However, neither H. cordata extract nor 2-undecanone had shown activity against C. fragariae despite its known activity against C. gloesporioides, which in turn motivates us in repositioning 2-undecanone as a selected candidate for Claisen-Schmidt condensation that enable access to several analogs (2a-f). Among the prepared analogs, (E)-1-(3-methylbenzo[b]thiophen-2-yl)dodec-1-en-3-one (2b) and (E)-1-(5-bromothiophen-2-yl)do- 34 dec-1-en-3-one (2f) showed promising activity against C. fragariae revealing a distinctive structural activity relationship (SAR). The generated analogs revealed a clear regioselectivity pattern through forming the C=C alkene bond at position C-1. These data open the window for further lead optimization and product development in the context of managing C. fragariae and S. invicta.

The work developed by the authors is very interesting, however, the study requires make minor and major corrections before being published. These corrections are shown in the attached document.

After making the indicated corrections, the article could be published in Molecules Journal.

Minor:

Comment 1. In lines 127, 131, 139, 148, 157, 166 and 175, compound numbers must be written in bold type.

Response 1. All compound numbers have been changed to bold type as suggested.

Comment 2. In lines 131, 139, 148, 157, 166 and 175, change (E) by (E)

Response 2. All (E) have been replaced by (E) as suggested.

Comment 3. In lines 106, 144, 145, 153, 162, 171 and 180, J (coupling constant) must be written in italics letters. 

Response 3. All J (coupling constant) have been changed to italics letters as suggested.

Comment 4. In lines 137, 145, 154, 163, 172 and 181, change CDCl3 by CDCl3 (must respect the subscripts)

Response 4. All CDCl3 have been replaced by CDCl3 as suggested.

Comment 5. In Figure 2 respect the double bond angle for compound 2 and indicate of carbon atoms numbering in structure:

Response 5. All double bond angle for compound 2 have been fixed and numbering of carbon atoms have been indicated in the structures as suggested.

Comment 5.1. Authors must assign carbon atoms numbering for all aromatic heterocyclic compounds part.

Response 5.1. Numbering of carbon atoms of all aromatic heterocyclic compounds part have been indicated in the structures as suggested.

Comment 6. In Supplementary Information, compound numbers must be written in bold type (in the index listing) and SI 2, SI 5, SI 8, SI 11, SI 14 and SI 17, check spaces for compound numbers (foot of spectra). 

Response 6. In Supplementary Information, all compound numbers have been changed to bold type (in the index listing) and SI 2, SI 5, SI 8, SI 11, SI 14, and SI 17, and spaces for compound numbers (foot of spectra) have been fixed as suggested.  

Major:

Comment 1. In lines 135, 143, 152, 161 and 179, what is p in (p, 2H)?

Response 1. “p” letter has been replaced with “m” to indicate multiple.

Comment 2. In line 135, the signal at d= 2.66 (t, 2H, H), the value of J must be included.

Response 2. In line 135, the signal at d= 2.66 (t, 2H, H), the value of J has been included as suggested.

Comment 3. In line 144, the signal at d= 2.66 (t, 2H, H), the value of J must be included.

Response 3. In line 144, the signal at d= 2.66 (t, 2H, H), the value of J has been included as suggested.

Comment 4. In lines 152 and 153, the signals at d= 2.71 (t, 2H, H) and 7.92 (d, 1H), the value of J must be included.

Response 4. In lines 152 and 153, the signals at d= 2.71 (t, 2H, H) and 7.92 (d, 1H), the values of J have been included as suggested.

Comment 5. In lines 161and 163, the signals at d= 2.69 (t, 2H) and 8.05 (d, 1H), the value of J must be included.

Response 5. In lines 161and 163, the signals at d= 2.69 (t, 2H) and 8.05 (d, 1H), the values of J have been included as suggested.

Comment 6. In line 170, the signal at d= 2.66 (t, 2H), the value of J must be included.

Response 6. In line 170, the signal at d= 2.66 (t, 2H, H), the value of J has been included as suggested.

Comment 7. In line 179, the signal at d= 2.59 (t, 2H, H), the value of J must be included.

Response 7. In line 179, the signal at d= 2.59 (t, 2H, H), the value of J has been included as suggested.

Comment 8. As I understand it, the synthesized compounds 2a-2f appear to be new. Therefore, authors should carry out full assignment of 1H and 13C NMR signals. This assignment must be made considering the numbering of carbon atoms in each synthesized compound (consider the carbon numbering in the figure above). For example:

(E)-1-(benzo[b]thiophen-2-yl)dodec-1-en-3-one (2a)… 1H and 13C NMR (400 MHz, CDCl3): δ =0.90 (m, 3H, H-12), 1.32 (m, 12H, H-6,7,8,9,10 and 11), 1.70 (p?, 2H, H-5), 2.66 (t, J = 7.4 Hz, 2H, H-4), 6.62 (d, J = 15.7 Hz, 1H, H-1)… etc; 13C-NMR (100 MHz, CDCl3): δ = 14.13 (C-12), 22.89 (C-11)…etc.

Response 8. All signals have been assigned as suggested.

Comment 9. Additionally, HRMS should be included for each new compound. 

Response 9. HRMS for all the synthesized compounds have been carried out and included as suggested.

Comment 10. Finally, the authors must describe some conclusions of the work carried out.

Response 10. The conclusion part has been implemented as suggested.

Reviewer 4 Report

In the present manuscript, authors have synthesized the regioselective analogs of 2-Undecanone obtained from Houttuynia cordata Thunb plant. The activity of the synthesized compounds has been tested against Solenopsis invicta (an invasive fire ant) and Colletotrichum fragariae (a phytopathogen). 2-Undecanone is a phytochemical well reported for its antifungal activity but it showed no activity against Colletotrichum fragariae  (a fungus). Furthermore the extract of the Houttuynia cordata Thunb plant (the source of 2-Undecanone for present study) was also found to be inactive against Colletotrichum fragariae. The research question was to synthsize analogs of 2-Undecanone that show antifungal activity against Colletotrichum fragariae.  From the synthesized analogs two compounds (2b & 2f) have shown reasonable activity against the Colletotrichum fragariae.In my opinion the results will definitely contribute to existing knowledge of antifungal compounds.The subject is interesting and the results seem promising.   The manuscript may be accepted after few minor changes.

 1) Purities of reagents/chemicals are not mentioned in the “Experimental” section. Mention the purities along with supplier details of all the chemicals used in that section.

2) Why is the digging bioassay also performed with “DEET”? Authors should make it clear in the text.

3)  Numbering of the “Supplementary Figures” should also appear in the main text at appropriate places.

4) Manuscript title and author’s details should also be provided in the supplementary information.

5) 2-3 more recent references (preferably from 2023) may be added.

6) Conclusion is missing. Authors must add a comprehensive “conclusion section” at the end of the manuscript.

Author Response

Reviewer 4:

Comments and Suggestions for Authors

In the present manuscript, authors have synthesized the regioselective analogs of 2-Undecanone obtained from Houttuynia cordata Thunb plant. The activity of the synthesized compounds has been tested against Solenopsis invicta (an invasive fire ant) and Colletotrichum fragariae (a phytopathogen). 2-Undecanone is a phytochemical well reported for its antifungal activity but it showed no activity against Colletotrichum fragariae (a fungus)Furthermore, the extract of the Houttuynia cordata Thunb plant (the source of 2-Undecanone for present study) was also found to be inactive against Colletotrichum fragariae. The research question was to synthesize analogs of 2-Undecanone that show antifungal activity against Colletotrichum fragariae.  From the synthesized analogs two compounds (2b & 2f) have shown reasonable activity against the Colletotrichum fragariae.In my opinion the results will definitely contribute to existing knowledge of antifungal compounds. The subject is interesting and the results seem promising.  The manuscript may be accepted after few minor changes.

Comment 1. Purities of reagents/chemicals are not mentioned in the “Experimental” section. Mention the purities along with supplier details of all the chemicals used in that section.

Response 1. The purities of the used chemicals as well as the supplier details were added to the experimental section as suggested.

Comment 2. Why is the digging bioassay also performed with “DEET”? Authors should make it clear in the text.

Response 2. DEET (N,N-diethyl-meta-toluamide) were tested. DEET was tested as a positive control for comparison whereas ethanol was included as negative control.

Comment 3. Numbering of the “Supplementary Figures” should also appear in the main text at appropriate places.

Response 3. Numbering of the “Supplementary Figures” has been implemented in the main text as suggested.

Comment 4. Manuscript title and author’s details should also be provided in the supplementary information.

Response 4. The manuscript title and authors’ details have been provided in the supplementary information as suggested.

Comment 5. 2-3 more recent references (preferably from 2023) may be added.

Response 5. We appreciate the reviewer’s point of view and would like to highlight that we have many recent references between 2020-2023 included such as reference number 2, 3, 10, 11, 21, 23, 26, 27, 28, 29, 35, and 36. 

Comment 6. Conclusion is missing. Authors must add a comprehensive “conclusion section” at the end of the manuscript.

Response 6. The conclusion part has been implemented as suggested.

Reviewer 5 Report

The article presents a very important and real problem, which emphasizes the usefulness of the collected data and their scientific significance. He believes that the introduction introduces the topic in a good way and brings the issues closer. Graphs and figures are clear and legible. I have only a minor remark regarding the discussion, which should be expanded a bit, as well as the bibliography. I did not find the conclusions section in the work, they should be underlined. I believe that if the authors make these changes, the article is suitable for publication.

Author Response

Reviewer 5:

Comments and Suggestions for Authors

Comment 1. The article presents a very important and real problem, which emphasizes the usefulness of the collected data and their scientific significance. He believes that the introduction introduces the topic in a good way and brings the issues closer. Graphs and figures are clear and legible. I have only a minor remark regarding the discussion, which should be expanded a bit, as well as the bibliography. I did not find the conclusions section in the work, they should be underlined. I believe that if the authors make these changes, the article is suitable for publication.

Response 1. We valued the respected reviewer’s appreciation of the presented work and the conclusion part has been now implemented as suggested.

Round 2

Reviewer 3 Report

I thank of paper authors for making all the suggested corrections by this referee.

The article is now presented in a clearer and more complete form.